# Mothers’ Awareness of the Correlation between Their Own and Their Children’s Oral Health

**DOI:** 10.3390/ijerph192214967

**Published:** 2022-11-14

**Authors:** Francesco Saverio Ludovichetti, Andrea Zuccon, Patrizia Lucchi, Giulia Cattaruzza, Nicoletta Zerman, Edoardo Stellini, Sergio Mazzoleni

**Affiliations:** 1Department of Neurosciences—Dentistry Section, Università degli Studi di Padova, 35128 Padova, Italy; 2Department of Pediatric Dentisrty, Università degli Studi di Verona, 37129 Verona, Italy

**Keywords:** pregnancy, infant oral health care, knowledge, mothers

## Abstract

Pregnancy is a critical time for a woman since it involves a series of changes in the body due to hormonal fluctuations and changes that can also influence the oral cavity and can lead to certain pathologies, such as carious lesions. Furthermore, it has been demonstrated over the years that having poor dental health during pregnancy might have a significant impact on the child’s oral health. The aim of the study is to analyze the level of knowledge and information of mothers on the importance of oral health care before, during, and after pregnancy as a preventive factor for the future oral health of the child. A questionnaire of 13 multiple choice questions was formulated and delivered from 24 February 2022 to 13 July 2022, aimed at women with at least one child. The Department of Pediatric Dentistry of the Borgo Cavalli Clinic in Treviso and the Maxillofacial Surgery Department of Santa Maria di Ca’ Foncello Hospital in Treviso were the data collection centers. Crosstabs with dependency variables were used to statistically analyze the results. The study cohort consists of 411 women, whose responses showed that the majority did not have dental visits before and during pregnancy, which is why 74% of the whole sample did not request or receive information from professionals. Most mothers knew that they had to cleanse their child’s mouth and, among them, those who responded more correctly, that is, who started before the eruption of the teeth, had received instructions from professionals. However, the low frequency of mothers performing dental checks during pregnancy did not allow mothers to become aware of the possibility of transmitting the carious lesions to their child, as conscious mothers represent 21%. Education from dentist and dental hygienists, gynecologists, and pediatricians is essential since they have an impact on mothers’ habits and behaviors and ensure favorable oral health for both the mother and the child.

## 1. Introduction

Pregnancy causes physiological changes in all of the maternal organs and systems, which return to normal after childbirth. These modifications affect the cardiovascular, hematological, gastrointestinal, renal, hepatobiliary, endocrine, respiratory and cutaneous systems, as well as the oral cavity [1,2]. Changes in hormone levels, such as estrogen and progesterone, which are associated with changes in oral hygiene habits and lifestyle, can make pregnant women more susceptible to oral cavity problems, which can have a negative impact on the child’s oral health [3,4,5]. Several oral lesions were reported as common during this period, with a special focus on a higher prevalence of dental caries and erosions, pregnancy tumors, and periodontal tissue problems, such as gingivitis and periodontitis [6]. 

Many women are not aware of the effects poor oral hygiene can have on them, on their unborn child, and on their pregnancy. Indeed, their knowledge about the transmission of oral pathogens, their diet, and oral hygiene habits are relevant in the etiology of dental caries in children and are important for the prevention of common oral diseases [7]. The main bacterium that causes human dental caries, Streptococcus mutans, plays a fundamental role in the etiology of this disease and its transmission is called “vertical” when it transfers from mother to child, for example, when sharing domestic utensils, during mouth-to-mouth feeding or, generally, whenever there is a passage of saliva from adult to the child’s mouth.

Mothers with poor oral hygiene and high levels of cariogenic bacteria raise the risk of infection in their newborns, who may therefore be more susceptible to developing early childhood caries [8]. Prevention of oral health and treatment of oral diseases before and during pregnancy improve the woman’s quality of life but they also reduce the evolution of dental diseases in the child and immediately promote better oral health for the child’s adult life [6]. The purpose of this study was to analyze mothers’ level of knowledge and information about the importance of oral care before, during, and after pregnancy as a preventive factor for the child’s future oral health, that is, to evaluate mothers’ awareness of the concept of “vertical transmission” from mother to child and of the bacteria that cause dental caries.

## 2. Materials and Methods

The study was conducted through a questionnaire of 13 multiple choice questions, which was delivered to women with at least one child from 24 February 2022 to 13 July 2022. The questionnaire was prepared using source of theory and research. It consists of general questions about mothers and their oral hygiene and that of their children, frequency of dental visits before and during pregnancy, and knowledge of the possibility of transmission of the carious pathology from mother to child. The topics were chosen from previously published studies. The Department of Pediatric Dentistry of the Borgo Cavalli Clinic in Treviso and the Maxillofacial Surgery Department of Santa Maria di Ca’ Foncello Hospital in Treviso were the data collection centers where mothers were bringing their children for dental visits or treatment. Ethical approval was waived by the Ethics Committee. Each participant signed an informed consent form as well as a request for permission to use the data.

The total obtained convenient sample was for 411 participants, to whom (during the delivery of the questionnaire) the content of the work and the methods of using the data were explained and the complete confidentiality of the information was also guaranteed. Statistical analysis was performed, allowing us to cross several variables in order to obtain more complete results.

## 3. Results

Our study consists of 411 subjects, all women with at least one child; 46% of the participants said they have two children, 30% only one, and 24% at least three or more, and the most prevalent age group of mothers is >35 years (58%). In terms of mothers’ oral hygiene habits, 54% brush their teeth twice a day; while regarding their children’s oral hygiene, 59% clean their child’s oral cavity twice a day but 48% only start cleaning it when their teeth start erupting. In total, 41% of women took their child to the dentist for the first time after the age of two and 39% when they noticed a problem, whereas only 16% and 4% went to the dentist at 6–12 months and 1–2 years, respectively. In the six months preceding and during pregnancy, 59% and 78% of mothers, respectively, said they did not have dental visits and only 26% of women asked for advice or received information from a dentist or dental hygienist. On the other hand, 61% believe that performing dental treatments during pregnancy is safe. In addition, they are not aware of the transmissibility of carious lesions (53%) and 42% believe that dental caries is not an infectious disease.

A statistical analysis was then performed through the chi-squared test to evaluate the presence of an association between the variables obtained from the questions in the questionnaire and eight statistical tables were obtained.

Table 1 contains the variable “Educational level” and the question “How many times a day do you brush your teeth?”. In this table we can see that most of the mothers have a high school diploma and among them, 141 brush their teeth twice a day. In addition, those who brush their teeth twice a day represent 54% of the total, while 29% (121) do so three times a day. Therefore, there is a correlation between the level of education and the number of times mothers brush their teeth during the day.

Table 2, Table 3, Table 4, Table 5 and Table 6 show the frequency observed in some variables, considering the question “Did you ask for advice or have you been given information by the dentist or the dental hygienist on the precautions to be taken during pregnancy?”.

Therefore, in Table 2, we have the variable “Did you have regular dental visits to the dentist during your pregnancy?” and approximately 67% of pregnant women did not have regular visits and did not receive information or professional advice on the care to be taken during gestation. It was observed that the idea of going to the dentist, whether regularly or not, influences the pregnant woman to receive more or less information on care.

A total of 252 of the interviewees believe that dental care is safe during pregnancy and, among these, 164 did not receive information on the treatments to be taken during pregnancy. Therefore, there is a relationship between considering dental treatments safe or unsafe for mother and child during gestation and asking for or receiving information from the dentist or dental hygienist about treatments to be taken during pregnancy (Table 3).

In all, 48% of women start brushing their child’s teeth only after the teeth have erupted (48%). Among them, 144 said they had not received any information. In addition, of all those who received instructions from a dentist or dental hygienist, 53 said they brush their child’s teeth once their teeth erupt and 33 said they clean their child’s oral cavity before their teeth erupt. Thus, receiving advice from the dentist or dental hygienist influences the age at which mothers begin brushing their children’s teeth (Table 5).

Table 5 shows how the number of times a day that the mother considers necessary to clean the baby’s mouth is influenced by the fact that the mother has received instructions from the dentist or dental hygienist. A total of 58% of mothers reported brushing their child’s teeth twice a day, while only 6% of mothers (26) reported brushing their teeth only once a day.

A total of 168 of the interviewees who filled out the questionnaire stated that they took their child to the dentist for the first time after he was two years old, followed by 160 mothers who brought him when they noticed a problem.

Only 16 mothers took their child to the dentist between the ages of 6 and 12 months. Lastly, the *p*-Value was less than 5%, indicating that receiving information during pregnancy influences the decision of when to take the child to the dentist for the first time (Table 6).

Table 7 shows the variable “In your opinion, a child born to a woman who has poor oral hygiene and is predisposed to dental caries is more likely to develop it, too?”. Among those who did not have any dental visits during pregnancy, 94 believe that there is no higher probability that the child could develop dental caries. Hence, it was concluded that there is a correlation between believing that a child born to a woman who has poor oral hygiene who is predisposed to dental caries is more likely to also develop it and having regular dental visits during pregnancy.

Table 8 shows that the mother’s level of education influences believing or not believing that caries is an infectious disease. Among mothers with a high level of education, 25% said carious disease is an infectious disease. Considering all women, 172 said that caries is not an infectious disease and, among these women, the majority has a secondary education diploma.

## 4. Discussion

Mothers’ oral hygiene, particularly during pregnancy, is crucial because it has a direct impact on the oral health of their children [9]. The American Dental Association describes how preventive, diagnostic, and restorative dental procedures that promote favorable oral health and remove pathological disorders can help improving both mother and child’s oral health [10,11]. The oral cavity of a baby, immediately after birth, is usually sterile or has only a few bacteria, which are transmitted during childbirth. Among the many factors that contribute to the development of the oral cavity and the oral microbiome during childhood, we find behaviors that promote saliva exchange, mainly between mother and child [12,13]. In fact, carious disease is defined as an infectious, transmissible, chronic-degenerative disease, with a multifactorial etiology, and it is currently one of the most widespread pathologies, especially among the pediatric population; in fact, many studies report that more than 50% of children have carious lesions during primary dentition [2,3,4,5,6]. 

Mothers’ oral hygiene, especially during pregnancy, plays an important role as it can directly affect the oral health of their children [9]. The possible correlation between the educational level of mothers and the frequency of brushing was verified here and statistical significance was found, revealing that mothers who have a high school diploma or who have attended university brush their teeth two or three times a day. On the other hand, it is different for those with a lower level of education, who, as we have seen, tend to brush their teeth more frequently than once or twice a day. A similar result to our study was obtained in the survey conducted by Ganesh et al., which showed that 66% of pregnant women brush their teeth twice a day [14]. These results could be attributed to the fact that having a higher level of education arouses a greater interest in knowing and understanding other topics that deviate from one’s own interest and therefore they more likely they also have notions in the health sector.

A correlation was also discovered between requesting or receiving information from a dentist or dental hygienist and performing periodic visits during pregnancy; 67% of pregnant women did not perform regular visits or receive information from professionals on the treatments to be taken during gestation. In accordance with our study, in that of M.T. Lydon-Rochelle et al., 54% of women did not receive advice on how to take care of their oral cavity [15]. If these women do not have an examination during the nine months, the professionals cannot evaluate their oral health conditions and, consequently, instruct them on the changes that may occur during gestation and on the proper oral hygiene, including that of the future child [16]. This shows that women are not yet sufficiently sensitized on the usefulness of having dental examinations during pregnancy.

Considering dental treatments to be safe or unsafe for mother and child during gestation was also influenced by receiving information from a dentist or dental hygienist on the treatments to be taken during pregnancy. In all, 61% of respondents said that dental care during pregnancy is safe, in accordance with the results of the study by R. Aiuto et al., where 52% of the sample considered it safe to perform dental treatments during pregnancy; despite this, it was found that both pregnant women and mothers do not attend the dental clinic often enough but come for urgent reasons [17]. Pregnancy is not a pathology but a physiological state; therefore, during pregnancy, you can go to the dentist safely, with a few simple precautions, and with this study, we can see a greater awareness of mothers as they are beginning to know more and more that it is safe to perform dental treatments during pregnancy.

We saw that mothers know they had to clean their children’s mouths but we also noticed that receiving advice influences the age at which mothers began brushing their children’s teeth. To prevent early bacterial colonization, which can result in the development of early childhood caries, the oral cavity must be cleaned from birth. Infants of mothers who received prenatal oral health care had lower incidences of ECC and Streptococcus mutans, according to Jin Xiao et al. [18].

However, the survey by N. Kumar et al. is in contrast with our study as it is stated that 85% of parents and 81% of children brush their teeth only once a day [19]. This could be related to the fact that the study sample belongs to a rural background in India. Thus, an adequate understanding of the effects that an improper brushing frequency might have is necessary since brushing teeth only once a day does not allow for the elimination of bacterial plaque in an effective manner.

We looked at the connection between bringing a child to the dentist for the first time and receiving advice from professionals and saw that those who received indications from professionals, although they were in the minority, answered more correctly because a sizeable number of them answered correctly: between 6 and 12 months. A child learns any behavior by observing the patterns around him and effective oral health habits are expected to be transmitted and assimilated in such a way that they correctly affect him for the rest of his life, similar to any other habit (6). In contrast to our study, L. Gavic et al. found that 41.23% of pregnant women believe their baby should be taken to the dentist for the first time around the first year of life, while the majority (53.23%) believe the first visit should be made once all deciduous teeth have erupted [20]. This more correct result could be due to the fact that the average age of the sample of mothers in this study is 28 years and therefore is younger than ours, which shows a higher prevalence of mothers > 35 years. Nowadays, the youngest, also due to the internet and social networks, obtain greater access to much more information, including on oral health. For instance, according to the American Academy of Pediatric Dentistry, the first pediatric dental visit should occur between 6 and 12 months of age [21].

Our study demonstrated the importance of periodic visits during pregnancy because among those who performed them, a minority of 14% said they did not think it was possible. Meanwhile, among those who did not perform them, in this case, the minority were those who knew it. The study is in line with the results of Thomas A. et al. [9] and Chacko V. et al. [22], who found that 22.2% and 35.6% of participants, respectively, were aware of the possibility of cariogenic bacteria transmission from mother to child. Maternal oral health is the first line of defense for a child’s healthy oral cavity. Additionally, kissing on the mouth, using the same cutlery as the mother, and cleaning the pacifier in the mother’s mouth are all risky behaviors to avoid in order to prevent cavities [6].

Moreover, the results of the study by Lilian Rigo et al. agree with ours, as 93.1% of mothers who were aware that dental caries is a transmissible disease had received information on the precautions to be taken during pregnancy and they were also those with a higher level of education [23]. This shows, as initially stated, that a more educated mother is more inclined to acquire information and therefore will be more prepared to ensure favorable oral health for the child from the perinatal period to adulthood. Mothers must be instructed to limit possible negative outcomes within the oral cavity during pregnancy.

## 5. Conclusions

The conducted research allowed us to affirm the necessity of appropriate primary prevention, which must be transmitted in partnership with the healthcare professionals who assist the woman during pregnancy, as they can influence mothers’ habits and behaviors, most of whom are still unaware of the importance of their oral health as a predictive factor of their child’s oral health.

## Figures and Tables

**Table 1 ijerph-19-14967-t001:** Distribution according to brushing frequency.

Variables	How Many Times a Day Do You Brush Your Teeth?		
Educational Level	Once a Day	Twice a Day	Three Times or More a Day
**Other**	10	12	6
**Middle School**	15	22	6
**High School**	42	141	61
**Univerity**	1	47	48

Note: Chi-squared test = 50.31, df = 6, *p*-Value = 0.0000048.

**Table 2 ijerph-19-14967-t002:** Distribution according to information received by dental professionals.

Variables	Did You Ask for Advice or Have You Been Given Information by the Dentist, the Dental Hygienist on the Precautions to Be Taken during Pregnancy?	
Did You Have Regular Dental Visits to the Dentist during Your Pregnancy?	No	Yes
**No**	276	46
**Yes**	30	59

Note: Chi-squared test = 99.15, df = 1, *p*-Value = 0.0000002.

**Table 3 ijerph-19-14967-t003:** Distribution according to information received by dental professionals about safety of performing dental treatments.

Variables	Did You Ask for Advice or Have You Been Given Information by the Dentist, the Dental Hygienist on the Precautions to be Taken during Pregnancy?	
In Your Opinion, Is It Safe for the Mother and Her Baby to Perform Dental Treatments during Pregnancy?	No	Yes
**No**	32	3
**I don’t know**	110	14
**Yes**	164	88

Note: Chi-squared test = 30.19, df = 2, *p*-Value = 0.00002.

**Table 4 ijerph-19-14967-t004:** Distribution according to information received by dental professionals about the age of starting cleaning the child’s mouth.

Variables	Did You Ask for Advice or Have You Been Given Information by the Dentist, the Dental Hygienist on the Precautions to Be Taken during Pregnancy?	
At What Age Should You Start Cleaning Your Child’s Mouth?	No	Yes
**After the teeth come out**	144	53
**I don’t know**	34	4
**Before the teeth come out**	44	33
**When the child eats solid foods**	84	15

Note: Chi-squared test = 22.53, df = 3, *p*-Value = 0.000051.

**Table 5 ijerph-19-14967-t005:** Distribution according to information received by dental professionals about frequency of brushing.

Variables	Did You Ask for Advice or Have You Been Given Information by the Dentist, the Dental Hygienist on the Precautions to Be Taken during Pregnancy?	
In Your Opinion, How Many Times a Day Do You Have to Clean Your Child’s Mouth?	No	Yes
**Once a day**	24	2
**Twice a day**	187	53
**Three times or more a day**	95	50

Note: Chi-squared test = 11.958, df = 2, *p*-Value = 0.00253.

**Table 6 ijerph-19-14967-t006:** Distribution according to information received by dental professionals about the first dental visit.

Variables	Did You Ask for Advice or Have You Been Given Information by the Dentist, the Dental Hygienist on the Precautions to Be Taken During Pregnancy?	
When Should You Take Your Child to the Dentist for the First Time?	No	Yes
**6–12 months**	33	34
**1–2 years**	6	10
**After 2 years**	125	43
**When you notice a problem**	142	18

Note: Chi-squared test = 51.049, df = 3, *p*-Value = 0.0004.

**Table 7 ijerph-19-14967-t007:** Distribution according to information received by dental professionals about regular dental visits during pregnancy.

Variables	Did You Have Regular Dental Visits to the Dentist during Your Pregnancy?	
In Your Opinion, a Child Born to a Woman Who Has Poor Oral Hygiene and Is Predisposed to Dental Caries Is More Likely to Develop It, Too?	No	Yes
**No**	94	13
**I don’t know**	179	38
**Yes**	49	38

Note: Chi-squared test = 32.767, df = 2, *p*-Value = 0.000007.

**Table 8 ijerph-19-14967-t008:** Distribution according to information received by dental professionals about dental caries as infectious disease.

Variables	Is Dental Caries an Infectious Disease?		
Educational Level	No	I Don’t Know	Yes
**Other**	11	10	7
**Middle School**	23	17	3
**High School**	107	115	22
**University**	31	37	28

Note: Chi-squared test = 28.541, df = 6, *p*-Value = 0.00008.

## Data Availability

Not applicable.

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
