# Peer review of "Mothers’ Awareness of the Correlation between Their Own and Their Children’s Oral Health"

_ijerph, 2022, doi:10.3390/ijerph192214967_

Round 1
Reviewer 1 Report
The authors have an interesting paper with important findings. However, the abstract reads very differently from the paper -- with more awkward turns of phrase and the word "Backrounds" at the beginning. (I think the authors meant "background." I suggest deleting it altogether.
The finds lack context. We know the opinions and behavior of pregnant women/mothers but a casual reader does not know the extent to which these behaviors deviate from medical wisdom. For instance, when should parents take their infants to the dentist and start brushing their teeth?
I assume that these authors had their study reviewed by an Ethics board of some sort, even if it's not called "IRB." Please mention this.
Please have the manuscript reviewed by an editor who can suggest some improvements in the English.
Author Response
Dear Reviewer,
thank you very much for reviewing our paper.
Your precious suggestions helped us improving the quality of our article.
We addressed every single comment you made.
Please, find below the point-by-point corrections.
Once again, thank you very much
Best regards
The Authors
Reviewer 1
- Abstract was reformulated according to suggestions
- Thank you for this helpful consideration, we add this informations
- Information about the Ethic board has been added in the materials and methods section
Reviewer 2 Report
This study highlighted the importance of raising oral health awareness of pregnant women and it is a good research topic with scientific merit.
Introduction: need to define the concept of "vertical transmission", e.g., through the pregnancy delivery? through the maternal sharing of utensils or mouth-to-mouth feeding? because one of the aspects that you are interested in is the pregnant women's knowledge and preventive behavior in this study.
Material and methods:
Is this study approved by an ethical committee? If yes, please cite it in the manuscript.
The participants of this survey are convenient samples, which should be mentioned.
How the questionnaire has been drafted? Or did you use the previously validated questionnaire? The questionnaire is highly suggested to be attached as an appendix when resubmitting.
The software and method of statistical analysis should be amended.
Results: Pls simplify the interpretation, the detailed description of which test was used and the p-value could be removed because they have been presented in your table.
Table ii iii iv v vi could be combined.
Discussion: pls avoid stating your results repeatedly in this part. The significance of your study and findings similar or different from other studies could be elaborated more.
Author Response
Dear Reviewer,
thank you very much for reviewing our paper.
Your precious suggestions helped us improving the quality of our article.
We addressed every single comment you made.
Please, find below the point-by-point corrections.
Once again, thank you very much
Best regards
The Authors
Reviewer 2
- The concept of “vertical transmission” has been defined
- Information about the ethic committee has been added
- We mentioned in the “material and methods” section the fact that the partecipants are a convenient sample
- The questionnaire was designed according to the most recent published literature. We added the questionnaire as an appendix (appendix A)
- The statistical analysis was provided by a Medical Statistic
- Interpretation was simplified by following your precious advice, we deleted the p-value and which test was used
- We did modify the discussion part without stating our results repeatedly and elaborating more the significance of our study.
Reviewer 3 Report
Thank you for sharing this research study.
Please see the below suggestions to revise the manuscript.
Abstract
Line 37 – ‘only’ is not placed in the correct part of the sentence.
Line 38 – Line 43 need revising to provide a more succinct conclusion
Introduction
Line 58 – revise the sentence
Line 59 – is that knowledge or attitude?
Line 62 to 64 – revise the sentence so it reads better with the rest of the paragraph
Line 65 – it sounded like poor oral hygiene and high levels of cariogenic bacteria are two separate entities
Line 70 – is ‘Comprehend’ the correct wording?
Materials and Methods
Line 76 – It needs details of the multiple choice questions. Is it a validated questionnaire? Where did you adapt this questionnaire from? If the questionnaire is designed by the authors, have the questions been tested? What are your recruitment methods? Random? Convenient? Were the participants bringing children into the Paediatric Dentistry department for treatment?
Results
Line 86 – 139 – results are stated systematically table by table but can be a bit more concise in describing the results
Line 113 – it did not make sense ‘they brush their teeth before their teeth erupt’, to revise the sentence
Discussion
Line 164 – definition of dental caries needs a reference
Line 169-181 – should this information be in the ‘Results’ section?
Line 199 – ‘correct’ is not a suitable word used here
In the discussion section, the findings of the study were repeated. Discussing each table’s results individually is unnecessary, which makes the writing boring. I suggest rewriting this section.
Conclusion
Line 262 – ‘collaboration between many professional figures’ – this sounded outdated; revise this to use more current collaboration terminology.
Author Response
Dear Reviewer,
thank you very much for reviewing our paper.
Your precious suggestions helped us improving the quality of our article.
We addressed every single comment you made.
Please, find below the point-by-point corrections.
Once again, thank you very much
Best regards
The Authors
Reviewer 3
- “only” was deleted
- Line 38-43 have been revised
- The word “attitude” has been changed with the word “knowledge”
- Line 62-64 has been re-written
- The word “comprehend” has been deleted
- This suggestion has been made also by reviewer 2, we decided to attach an appendix (appendix a) with the questionnaire. The questionnaire was designed according to the most recent published literature; we stated in text that the sample was a convenient sample (as also requested by reviewer 2). We add the information stating the mothers were bringing their children both for dental visit or dental treatment in the data collection Centers.
- The “results” part has been simplified
- We change “brush their teeth” with “clean their oral cavity”
- Many references about dental caries are present in text
- We change the word “correct” with the word “proper”
- Discussion part was simplified
- “Collaboration between many professionals figures” has been change with “cooperation between health professionals”
Round 2
Reviewer 2 Report
Dear Authors,
Thanks for your work to revise the manuscript and most of the limitations being improved.
However, the presentation of the tables should be improved for the analysis of one dependent variable has never been presented separately in such many tables in a formal scientific report. you can search for the other paper and read how to present the results of the chi-square test. In addition, the "three lines" table has been always applied.
Table ii iii iv v vi could be combined.
Author Response
Dear Reviewer,
thank you very much for your considerations.
Please find here some papers already published where tables have been presented and published as in this one:
- Ludovichetti FS, Signoriello AG, Zuccon A, Padovani S, Mazzoleni S. The Role of Information in Dental Traumatology in Patients during Developmental Age: A Cognitive Investigation. Eur J Dent. 2022 May;16(2):296-301.
- Togoo RA, Al-Almai B, Al-Hamdi F, Huaylah SH, Althobati M, Alqarni S. Knowledge of Pregnant Women about Pregnancy Gingivitis and Children Oral Health. Eur J Dent. 2019 May;13(2):261-270.
-
Oral disorders and oral hygiene habits in pregnancy: Cognitive survey
Ludovichetti, F.S., Signoriello, A.G., Piccoli, V., ...Stellini, E., Mazzoleni, S.
Dental Cadmos, 2021, 89(10), pp. 778–788
Impact of COVID-19 Pandemic on Dental Education: Perception of Professors and Students
Costa, E.D., Brasil, D.M., Santaella, G.M., ...Ludovichetti, F.S., Freitas, D.Q.
Odovtos - International Journal of Dental Sciences, 2022, 24(1), pp. 122–133
Author Response
Dear Reviewer,
thank you very much for your considerations.
We addressed all of them and so, improved the quality of our paper.
Please, find below the corrections made.
Once again, thank you for your time
Best regards
The Authors
- Line 36 to 43 have been changed
- Line 80: we added informations about the questionnaire and attached appendix 1 (which is indeed the questionnaire)
- Line 113-115: corrected
- Line 117-120: corrected
- Line 128: corrected
- Results section has been modified
- Line 180 to 191 have been moved to the results section as suggested
- Conclusion has been changed